# Disentangled Face Representations in Deep Generative Models and the Human Brain

**Paul Soulos**
Johns Hopkins University
psoulos1@jhu.edu

**Leyla Isik**
Johns Hopkins University
lisik@jhu.edu

## Abstract

How does the human brain recognize faces and represent their many features? Despite decades of research, we still lack a thorough understanding of the computations carried out in face-selective regions of the human brain. Deep networks provide good match to neural data, but lack interpretability. Here we use a new class of deep generative models, disentangled representation learning models, which learn a latent space where each dimension "disentangles" a different interpretable dimension of faces, such as rotation, lighting, or hairstyle. We show that these disentangled networks are a good encoding model for human fMRI data. We further find that the latent dimensions in these models map onto non-overlapping regions in fMRI data, allowing us to "disentangle" different features such as 3D rotation, skin tone, and facial expression in the human brain. These methods provide an exciting alternative to standard "black box" deep learning methods, and have the potential to change the way we understand representations of visual processing in the human brain.

## 1 Introduction

Humans are very good at recognizing faces despite the complex high dimensional space that faces occupy. A given face can vary across many dimensions. Some dimensions (such as 3D rotation and lighting) are constantly changing and thus irrelevant to recognizing the face, while others (such as facial features or skin tone) are generally stable and useful for recognizing individuals, and still others (such as hair style) can change but also offer important clues for recognition. Face networks in the macaque and human brains have been thoroughly mapped and individual cells in the macaque have been identified that are selective to particular face features [1, 2]. However, the overall computations in face regions are still poorly understood. This lack of understanding can be seen in the relatively poor decoding of face identity from fMRI data compared to other visual categories [3].

Recently, deep generative models have been shown to provide a good match to human fMRI data and provide high decoding accuracy of individual faces [4]. These models, however, transform faces into a high dimensional latent space that is difficult to interpret, and thus provide only limited additional information about the underlying neural computations. Here we used a new class of deep generative models, disentangled representation learning models, to understand the computations underlying human face recognition. There are multiple disentangled representation learning models [5–9], and we will refer to all of the models based on Variational Autoencoders [10] as Disentangled Variational Autoencoders (dVAEs). dVAEs learn a latent space that "disentangles" different explanatory factors in the training distribution [11]. When applied to faces, these models isolate specific face features (such as lighting, viewpoint and facial expression) in specific neurons [5–9]. This latent space is computationally effective and highly interpretable by humans. We asked if the human brain might be using a similar disentangled feature space to recognize faces.

2nd Workshop on Shared Visual Representations in Human and Machine Intelligence (SVRHM), NeurIPS 2020.

We trained a dVAE to reconstruct face images through a bottleneck and confirmed that its latent space contains several interpretable features. We learned a linear transformation between the dVAE latent dimensions and face-selective voxels in the human brain, and used this transformation as an encoding model to predict the fMRI responses to held-out face images. We showed that this encoding model is as accurate as a standard VAE, suggesting that the added interpretability constraint on the dVAE does not affect its match to human neural data. Finally, we found evidence with the dVAE that key dimensions such as rotation, skin tone, smile, and gender appearance are also disentangled in the human brain.

## 2  Disentangled Variational Autoencoder

Variational Autoencoders (VAEs) [10] are a common framework for producing generative models with neural networks. A major goal of the machine learning community is to disentangle the factors of variation that underlie a dataset [11]. These explanatory factors can change independently and produce large changes in the data distribution while requiring only minor changes in a disentangled latent space. The generative capacity of VAEs make them a prime candidate for learning disentangled representations, and there are many methods that attempt to learn disentangled latent spaces through supervised training methods [5] and information-theoretic approaches [6–9]. We

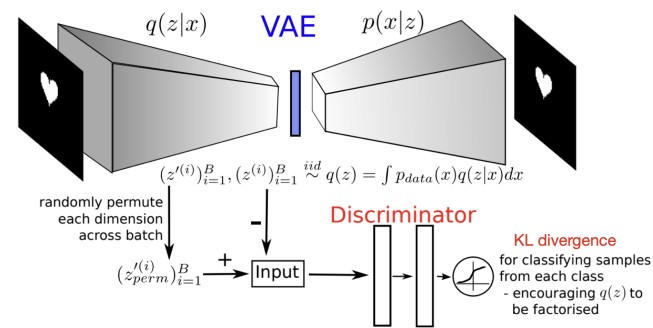

Figure 1: Diagram of the dVAE adapted from [9]. The dVAE has a standard VAE architecture with an additional loss term to encourage the elements of q(z) to be factorized. We learn a linear map between the VAE latent vector (highlighted in blue) and fMRI data.

use FactorVAE [9] as our model because it is a simple approach. In addition to the standard VAE objective, FactorVAE adds an additional term to the objective function which minimizes the Total Correlation [12]. The Total Correlation is the KL Divergence between a joint distribution of random variables and the product of those the individual random variables. Let $q(z)$ be our joint distribution of $d$ random variables and $\bar{q}(z) = \prod_{i=1}^{d} q(z_i)$ be the product of the individual variables. By minimizing the KL Divergence between $q(z)$ and $\bar{q}(z)$, FactorVAE encourages a latent space where the different dimensions are independent from each other.

We trained a standard VAE and a FactorVAE using the CelebA dataset [13] which consists of 200,000 face images. The two models both had latent vectors of size 24 and achieved similar image reconstruction loss (see Appendix A.1 for more training details). After training the FactorVAE model, two raters manually inspected the latent traversals and tagged the dimensions with the human interpretable transformations. Figure 2 shows the latent traversals for two of these dimensions representing 3D rotation and smile. Additional traversals for gender and skin tone are shown in Figure 6 in the Appendix.

## 3  fMRI analysis

### 3.1  fMRI data

We used publicly available fMRI data of four subjects from [4]. Briefly, subjects viewed around 8000 "training" face images each presented once, and 20 "test" face images presented between 40-60 times each. Subjects were scanned on separate face-object localizer runs, which we used to identify voxels that responded significantly more to faces than objects (p<10^-4, uncorrected). Data were pre-processed and projected onto subjects' individual cortical surfaces using Freesurfer [14].

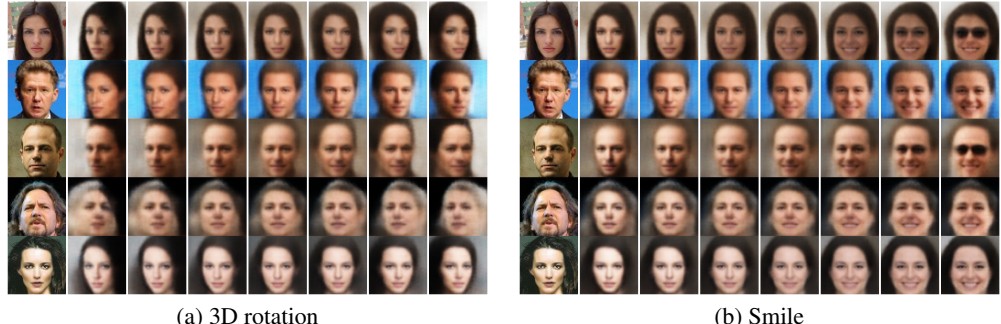

(a) 3D rotation           (b) Smile

Figure 2: dVAE latent dimensions. Examples of faces generated by altering the dVAE latent dimension representing (a) 3D rotation and (b) smile (note that smile is slightly entangled with sunglasses towards the positive side). Input face on left and face reconstructions for different values of the latent dimension shown on right (all other latent dimensions are kept fixed).

## 3.2 Encoding model

We estimated a linear map between the latent dimensions in our VAE and the fMRI data via a generalized linear model (GLM). To do this we extracted the latent representation for each training image, and used these weighted latent dimensions as regressors in the GLM. We also estimated the response to each test image as a separate regressor in the GLM.

To test the accuracy of the encoding model, we extracted the latent dimensions for each test image and multiplied this by the linear transformation learned in the GLM to get a predicted voxel response to each test image (Figure 3). We then correlated the predicted fMRI response in each voxel activity to the true voxel activity across all test images.

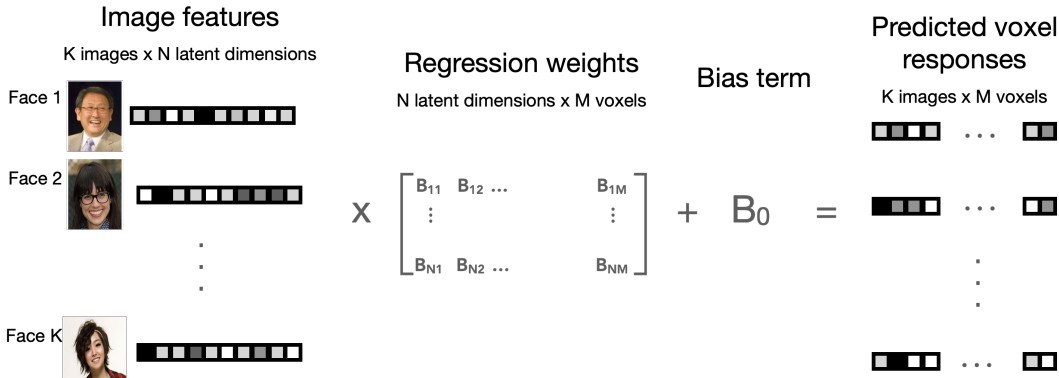

Figure 3: Encoding model procedure. Each test image is passed through the VAE encoder to extract its N-dimensional latent representation. This latent representation is multiplied by the regression weights learned from the training images via GLM. We next add a bias term coding for the presence of a face in the GLM. This produces a predicted voxel response for each test image. These predicted voxel responses are correlated with the true voxel responses to evaluate encoding performance.

## 4 Results

### 4.1 Encoding model performance

We compared the encoding model performance of our dVAE to the standard VAE. We found that the dVAE encoding results were comparable to a VAE, and led to better encoding performance in two of three subjects (Table 1, Figure 4). This suggests that dVAEs provide a similarly good match to human neural data as VAEs despite the additional disentanglement term applied during training.

Table 1: Average Spearman Correlation across face selective voxels. All correlation values are significantly above chance (p <0.001 based on a permutation test).

|      | Subject 1 | Subject 2 | Subject 3 | Subject 4 |
|------|-----------|-----------|-----------|-----------|
| dVAE | .071      | .165      | .023      | .118      |
| VAE  | .034      | .176      | .014      | .21       |

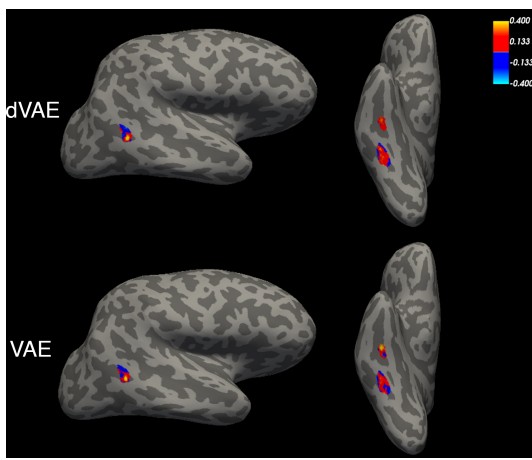

Figure 4: Correlation (Spearman's rho) for Subject 1 in face-selective voxels between the fMRI activity predicted by the dVAE (top row) and VAE (bottom row) and the ground-truth fMRI recordings on the test images.

## 4.2   Disentangling features in the human brain

The main promise of a disentangled model over standard deep learning models is the ability to use their disentangled representations to interpret brain data. To do this, we visualized the beta weights for two of the latent dimensions: rotation and smile. These were two of the 15 dimensions consistently labeled by both raters. Since the dVAE learns to de-correlate its latent dimensions, it is straightforward to examine each of their independent beta weights.

We found that these two dimensions are contained in largely non-overlapping voxels in all four subjects as seen in Figure 5 (though see Appendix Figure 7 for two additional dimensions with more overlapping voxels). This provides a promising first proof-of-concept that the variables disentangled by the dVAE may also be disentangled in the human brain. In addition to the spatial segregation, the different latent dimensions have a distinct weighting that is consistent across subjects.

## 5   Discussion

Here we showed that a dVAE serves as a good encoding model of face processing in the human brain, and that it allows us to "disentangle" two key dimensions of face processing: gender and rotation. This model provides, for the first time, an interpretable and straightforward way to identify the representations for several key face features in the human brain. Prior work on disentangled representations has shown that they are helpful for generalizing on downstream reasoning [15]. It is possible that the human brain may use disentangled representations to achieve robust generalization. If the human brain implements disentangled representations, different architectures and training objectives which encourage disentanglement could reveal new biologically plausible inductive biases.

Prior work has shown that discriminative deep neural networks provide a good match to human neural data for general visual processing [16]. More recent work though, suggests that generative models may provide a better match to face responses than discriminative models [17]. In particular, prior work showed high encoding/decoding performance of faces in human fMRI data [4]. This work was also able to provide some information about how different face features are coded in the brain since it

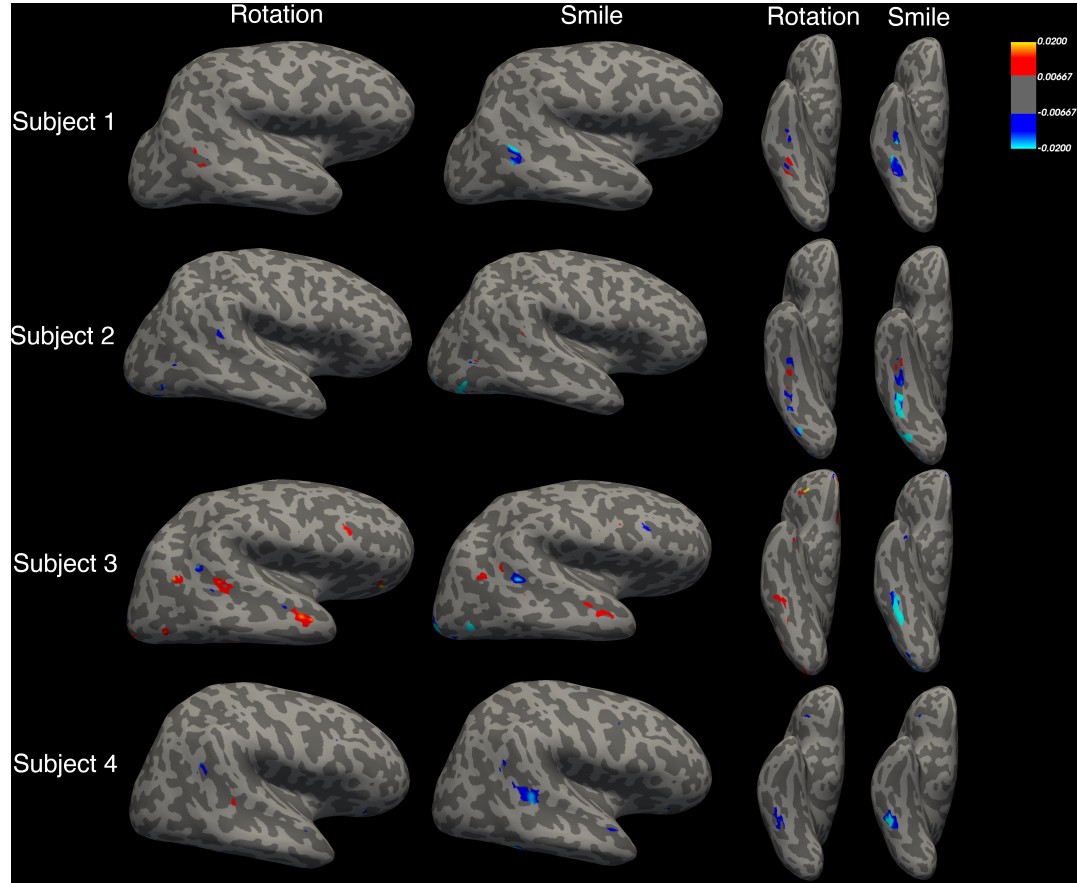

Figure 5: Beta weights in the right hemisphere across four subjects for two latent dimensions in the dVAE model for 3D rotation (columns 1 and 3) and smile (columns 2 and 4). The voxels corresponding to rotation and smile are largely non-overlapping. Additional betas for the gender appearance and skin tone disentangled dimensions can be seen in the Appendix Figure 7

.

used a dataset of face images that was annotated with information like gender appearance and facial expression.

Our work extends these results and provides several advantages over traditional "entangled" deep learning models. First, since the feature information is learned in the disentangled model, we do not need to rely on annotated datasets. Not only is annotation expensive, but some continuous features (like lighting or rotation changes) are almost impossible to label in uncontrolled natural images. In addition to a more thorough characterization of the representations of different disentangled features, dVAEs may also be useful for understanding transformation-invariant representations of faces. Several features "disentangled" by the dVAE are identity-preserving transformations, such as lighting and viewpoint. By discounting the contribution of these features, we can decode different face identities with less data and better understand how identity information is represented in the brain, despite the fact that identity is not a feature that is disentangled by the model.

One other study recently used dVAEs to analyze face monkey physiology data [18]. They found that different dimensions disentangled by the dVAE were highly correlated with single cell responses in the anterior face patch of monkeys. Here we showed that this work can be extended to whole-brain fMRI data. This allows us to move beyond single cell tuning and examine computational properties of the entire face processing network.

# 6 Broader Impact

Computational models of face perception have many potential adverse societal affects, including invasion of privacy and racial bias. Since disentangled models can isolate variable that we know lead to bias (such as skin tone, gender appearance, and hair style), they may further these problems. On the other hand, if applied thoughtfully, disentangled models can be used as a tool to combat bias in downstream decision making, since they make it easier to identify and discount potential sources of bias. Indeed, prior work has shown that disentangled representations can increase the fairness on downstream prediction tasks [19]. Purely algorithmic attempts to correct racial bias in the computational sciences, however, are sure to fall short [20]. Therefore we are also committed to examine racial bias in our training set and model by comparing model reconstruction performance on different races, and addressing any biases this analysis reveals. Another pervasive issue in face recognition research is "gender" classification in face images [20]. This is problematic first because it treats gender as a binary dimension and second because it assumes gender can be identified based purely on visual facial appearance. While we identified a latent variable that seemingly coded for stereotypically "masculine" vs. "feminine" facial features (referred to as "gender appearance" above), we do not claim the model can "classify gender".

# 7 Conclusion

Here we showed that disentangled deep generative models provide a good match to human neural data, and improve interpretability without sacrificing performance. These models allow us to identify human-relevant and computationally important face features in the brain, and are a promising tool to understand the complex visual representations underlying human face perception.

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

# A   Appendix

## A.1   Model training details

We trained our models using the Disentanglement Lib framework [21]. The CelebA input images were first scaled to 64x64x3. The first four layers are convolutional layers. The first two layers have 32 filters, a kernel size of 4, a stride of 2, and use the ReLU activation function. The third and fourth layer have 64 filters, a kernel size of 2, a stride of 2, and use the ReLU activation function. This is followed fully-connected layer to dimension 256 using the ReLU activation function. Finally, this last vector is linearly projected to two vectors of the latent dimension size to represent the mean and variance of the VAE. The decoder follows a similar architecture, but in reverse with deconvolutional layers.

We perform a hyperparameter search over the size of the latent vector and the hyperparameter coefficient $\gamma$ applied to the disentanglement objective. We searched across latent vector sizes of 24 and 32, and searched for $\gamma$ values in [3, 6.4, 12, 24]. For every hyperparameter combination, we train two models to avoid one model getting stuck in a bad local optimum. We qualitatively analyzed the models to see which one was disetangling well. The chosen model used a latent dimension of 24 and $\gamma$ of 24.

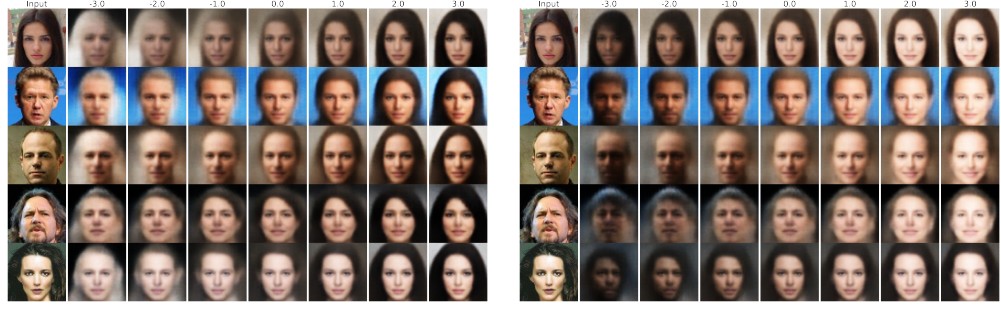

| (a) Gender appearance | (b) Skin tone |
| --- | --- |

Figure 6: dVAE latent dimensions. Examples of faces generated by altering the dVAE latent dimension representing (a) gender appearance and (b) skin tone. Input face on left and face reconstructions for different values of the latent dimension shown on right (all other latent dimensions are kept fixed).

After the FactorVAE model was trained, we trained a VAE with the exact same hyperparameters. This is akin to trainng another FactorVAE model where the Total Correlation coefficient hyperparameter $\gamma$ is set to 0.

The final FactorVAE achieves a reconstruction loss on the test set of 6477 while the final VAE achieves a reconstruction loss of 6470.

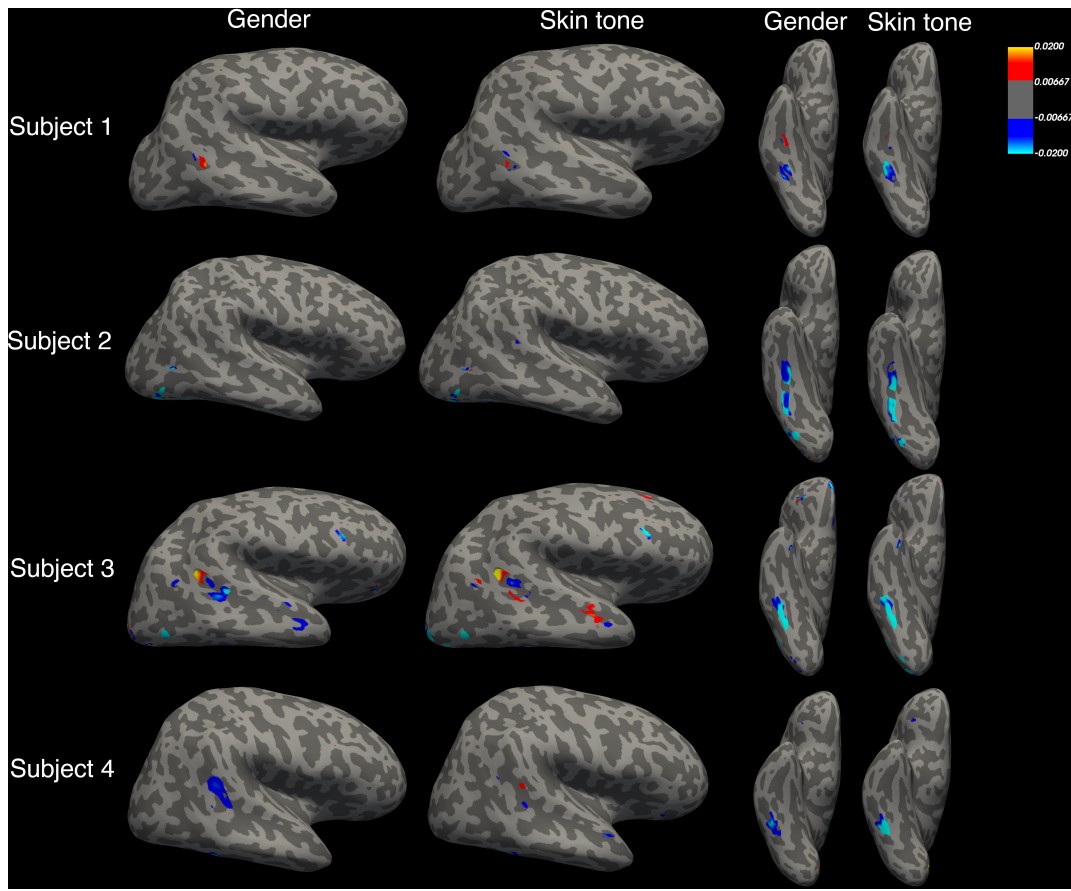

Figure 7: Beta weights in the right hemisphere across four subjects for two latent dimensions in the dVAE model for gender appearance (columns 1 and 3) and skin tone (columns 2 and 4).

