# OpenReview forum: "Disentangled Face Representations in Deep Generative Models and the Human Brain"
_NeurIPS.cc/2020/Workshop/SVRHM — SVRHM@NeurIPS Poster_

### Official Review · AnonReviewer2 · 2020-10-27
**interesting idea with limited results**

**Rating:** 6
**Confidence:** 3

**Review:**

This paper is investigating VAEs and disentangled VAEs as potential representation of faces in the human brain. It claims that disentangled VAEs similarly well match fmri data as VAEs but have the benefit of additional control and disentanglement. They further claim that gender and 3D rotation (yaw) is represented disentangled in the brain.

I'm not an expert of evaluation of fmri data, but I have seen such evaluations before. And for me those results are just not convincing to me. The statistical evidence is rather limited and the effect is also not really consistend among the 3 subjects. It is also unclear why not a VAE-GAN as proposed in the paper the provides the dataset was used for comparison?

I got confused with the wording of dVAE and FactorVAE, that is refeRring to the same, right? In the discussion you then refer to [18] which uses a beta-VAE, which is different (?) but say they use a dVAE?

Overall the paper is interesting and might hold interesting results, but for me it is not conclusive. I think that is fine for a workshop paper, but I think the limited results are overstated in this work

---

### Official Review · AnonReviewer3 · 2020-10-28
**Results need more work to validate the claims, a good step towards interpretable encoding models**

**Rating:** 6
**Confidence:** 5

**Review:**

Summary of contributions:
The authors propose a disentangled VAE as a more interpretable encoding model of human fMRI activity. They show that dVAE shows better encoding than a simple VAE and demonstrate that it can be used to disentangle regions in the brain corresponding to meaningful interpretable dimensions such as 3D rotation and gender

Significance:
The approach is relevant to understanding how the human brain performs face perception. However, the results do not provide sufficient evidence and might need some improvement.

Pros:
1. dVAE seems to find interpretable latent such as Gender and 3D rotation
2. The paper is written clearly

Cons:
1. In Table 1, the Spearman correlation is quite low and might even not be significant
	1. I would encourage authors to perform some statistical test to find whether the correlations are significant or not
	2. The authors might consider finding out data quality using noise ceiling or split-half reliability for the 20 test faces to estimate data quality and compare the correlation with noise ceiling
2. In Figure 4, the results can be visualized in a better way showing both the beta-weights in a single brain map to find out whether there is an overlap or not. One possibility is showing beta weights corresponding to gender and rotation in a single brain map using different colormaps. Further, the authors might consider performing statistical tests to validate their claim

Suggestion to authors:
Please read cons

---

### Official Review · AnonReviewer1 · 2020-10-29

**Rating:** 7
**Confidence:** 3

**Review:**

The paper shows that the disentangled variational autoencoder models for faces learn a set of latent dimensions which are shared with the representation in the human brain. Their main results are 1) the average encoding performance for dVAE is comparable to that for regular VAE. 2) Latent dimensions in their model correlate with non-overlapping regions in the brain.

Overall, I believe this paper demonstrates interesting findings which suggest that compared to standard deep learning models, dVAE models can potentially provide additional interpretation of the representation in the human brain.

Pros:
- Using VAEs for modeling visual processing in the human brain is interesting.
- Some latent variables in dVAE are easily interpretable.
- Clear delivery

Cons:
- It is not clear why there is a distinction between training and test face images in line 80.
- I see that only two of the latent variables were included in analysis for the results in section 4.2. For the other latent variables, do they map to voxels that spatially overlap with others? Those latent variables may be less human interpretable, but it would still be helpful to understand whether the human brain and dVAE have additional shared disentangled features.
- A quantification on the overlap of voxels correlated with the latent variables will be helpful.
- How does the number of latent variables in dVAE affect your results?

---

### Public Comment · ~Paul_Soulos1 · 2020-12-08
**Response to reviewers**

Thank you to all three reviewers for your insightful and helpful comments. We used your suggestions to improve our paper in several ways. We were able to resolve technical issues and added the fourth subject in the fMRI dataset that we use (the previous Subject 2 is now Subject 3, and the previous Subject 3 is now Subject 4). There was some concern with the low correlation values that we showed in Table 1 as well as a lack of statistical tests. We made two changes to address these concerns. First, we added bias terms to our encoding model that correspond to the presence of a face. This technique was originally used in Rullen and VanReddy 2019 whose dataset we use, and improves our correlation values. Second, we ran non-parametric statistical tests and found significant correlations (p<0.001) for all subjects. We also added two additional disentangled dimensions, smile and skin tone, to our analyses. For space reasons, some results were moved to the supplemental material.

In the future we plan to incorporate your additional feedback, including quantifying the voxel overlap (or spatial disentanglement) between all pairs of latent dimensions, and comparison with other models such as VAE-GAN, standard autoencoders, and discriminative models.

---

### Decision · Program_Chairs · 2020-11-02

Accept (Poster)